# Role of Tobramycin in the Induction and Maintenance of Viable but Non-Culturable *Pseudomonas aeruginosa* in an In Vitro Biofilm Model

**DOI:** 10.3390/antibiotics9070399

**Published:** 2020-07-10

**Authors:** Gianmarco Mangiaterra, Nicholas Cedraro, Salvatore Vaiasicca, Barbara Citterio, Roberta Galeazzi, Emiliano Laudadio, Giovanna Mobbili, Cristina Minnelli, Davide Bizzaro, Francesca Biavasco

**Affiliations:** 1Department of Life and Environmental Sciences, Polytechnic University of Marche, via Brecce Bianche, 60131 Ancona, Italy; n.cedraro@pm.univpm.it (N.C.); s.vaiasicca@pm.univpm.it (S.V.); r.galeazzi@staff.univpm.it (R.G.); g.mobbili@staff.univpm.it (G.M.); c.minnelli@pm.univpm.it (C.M.); d.bizzaro@staff.univpm.it (D.B.); f.biavasco@staff.univpm.it (F.B.); 2Department of Biomolecular Science, Biotechnology Section, University of Urbino “Carlo Bo”, via Arco d’Augusto 2, 61032 Fano, Italy; barbara.citterio@uniurb.it; 3Department of Materials, Environmental Sciences and Urban Planning, Polytechnic University of Marche, via Brecce Bianche, 60131 Ancona, Italy; e.laudadio@staff.univpm.it

**Keywords:** cystic fibrosis, *Pseudomonas aeruginosa* biofilms, viable but non-culturable forms, tobramycin, recurrent infections

## Abstract

The recurrence of *Pseudomonas aeruginosa* (PA) biofilm infections is a major issue in cystic fibrosis (CF) patients. A pivotal role is played by the presence of antibiotic-unresponsive persisters and/or viable but non-culturable (VBNC) forms, whose development might be favored by subinhibitory antibiotic concentrations. The involvement of tobramycin and ciprofloxacin, widely used to treat CF PA lung infections, in the abundance of VBNC cells was investigated in PA biofilms models. In vitro biofilms of the laboratory strain PAO1-N and the clinical strain C24 were developed and starved by subculture for 170 days in a non-nutrient (NN) broth, unsupplemented or supplemented with one-quarter minimal inhibitory concentration (MIC) of tobramycin or ciprofloxacin. VBNC cells abundance, estimated as the difference between total live (detected by qPCR and flow cytometry) and colony forming unit (CFU) counts, showed a strain- and drug-specific pattern. A greater and earlier abundance of VBNC PAO1-N cells was detected in all conditions. Exposure of the C24 strain to NN and NN + ciprofloxacin induced only a transient VBNC subpopulation, which was more abundant and stable until the end of the experiment in tobramycin-exposed biofilms. The same response to tobramycin was observed in the PAO1-N strain. These findings suggest that low tobramycin concentrations might contribute to PA infection recurrence by favoring the development of VBNC forms.

## 1. Introduction

*Pseudomonas aeruginosa* causes chronic infections [1], which are the main cause of death in cystic fibrosis (CF) patients. Biofilm production [2], multidrug resistance, the formation of pathogen reservoirs in different regions of the lung [3] and persistent, dormant bacterial forms [4] are key factors in infection resilience. Viable but non-culturable (VBNC) cells, a later stage of dormancy, do not grow on bacteriological media, and exposure to favorable conditions is not sufficient for the immediate recovery of full metabolic activity and culturability; for this to happen, conditions must include factors such as those protecting against oxidative stress [5]. VBNC cells have been demonstrated in CF sputum [6,7,8], being more abundant in patients showing chronic *P. aeruginosa* colonization and limited improvement in lung function after treatment [6]. Our group has documented VBNC cells in a study of clinical samples, where *P. aeruginosa* abundance was measured by routine culture-based methods and by a species-specific qPCR protocol, which was demonstrated to detect both culturable and non-culturable viable bacteria [8]. Although the factors involved in VBNC cell induction in the lungs of CF patients are still to be elucidated, their pulmonary environment is characterized by the presence of multiple stress factors that include low-level oxygen, catabolic waste, nutrient depletion, suboptimal pH as well as antibiotics, since chronic infections are repeatedly treated with drugs such as tobramycin and ciprofloxacin. Notably, the antibiotic may be found in subinhibitory concentrations in the biofilm matrix due either to its poor penetration or to its decrease between two different therapeutic cycles [9]. This is a cause of concern, particularly in strong biofilm-producing bacteria like *P. aeruginosa*. According to the hormesis theory, developed by Davies and co-workers, subinhibitory antibiotic concentrations may modulate bacterial gene expression [10]. We have previously demonstrated a role for vancomycin and quinupristin/dalfopristin in the maintenance of VBNC *Staphylococcus aureus* cells in biofilms cultured in nutrient-depleted medium [11]. According to a more recent study [12], culture in the presence of gentamicin resulted in a more robust induction of VBNC *P. aeruginosa* cells than in the presence of carbenicillin or colistin, suggesting a main role for the protein synthesis inhibitors. This work was undertaken to assess the ability of subinhibitory concentrations of tobramycin or ciprofloxacin—which are routinely used to treat CF patients with *P. aeruginosa* lung infections [13,14]—to induce the development of VBNC cells in in vitro biofilms. Total live cells were detected by the developed qPCR protocol proven to reliably detect all viable (i.e., culturable and non-culturable) *P. aeruginosa* cells [8].

## 2. Results

### 2.1. Assessment of Antibiotic Susceptibility

The susceptibility of *P. aeruginosa* strains PAO1-N and C24 to ciprofloxacin and tobramycin was evaluated by minimal inhibitory concentration (MIC) determination. Strain PAO1-N was susceptible to both antibiotics, with MIC values of 0.25 mg/L and 1 mg/L, respectively, whereas strain C24 showed low-level resistance with MIC values of 4 and 32 mg/L, respectively.

### 2.2. Stress Exposure of P. aeruginosa Biofilms and VBNC Cells Detection

PAO1-N and C24 biofilms were developed in vitro in rich medium and then starved by immersion in non-nutrient medium (NN), in NN + 1/4 MIC ciprofloxacin (0.062 mg/L and 1 mg/L, respectively) or in NN + 1/4 MIC tobramycin (0.25 mg/L and 8 mg/L, respectively). Biofilms were monitored for bacterial viability (qPCR counts) and culturability (colony forming unit (CFU) counts) for 170 days at approximately 15-day intervals, starting 60 days from stress exposure. More than 92% of the biofilm matrix of both *P. aeruginosa* strains was consistently recovered, demonstrating the reliability of biofilm detachment from the petri dishes. Moreover, to confirm the reliability of qPCR in the detection of viable cells only, *P. aeruginosa* PAO1-N total viable cell (TVC) counts were performed by both qPCR and live/dead flow cytometry (FC) assays (Figure 1); similar live bacterial counts were always obtained, confirming the ability of the qPCR protocol to exclude extracellular DNA (eDNA) (Appendix A).

Culturable cells of both strains were recovered at all time points. However, the two strains showed different amounts of TVCs and culturable cell counts over time (Figure 2).

Strain C24 showed a progressively lower number of both TVCs and culturable cells compared to the counts of day 0, i.e., immediately before stress exposure. A significant difference between total viable and culturable cells (reflecting the presence of VBNC forms) was detected only in antibiotic-exposed biofilms, with a marked difference between ciprofloxacin- and tobramycin-exposed biofilms. In fact, the former biofilms contained a considerable amount (about 5 × 10^6^) of VBNC cells, which, however, were no longer detectable at the end of the experiment, whereas in the latter biofilms, VBNC forms were consistently observed until the end of the experiment.

A large amount of VBNC forms, ranging from 85% to 97% of TVCs (Table 1), were counted in all *P. aeruginosa* PAO1-N biofilms since the first time point (60 days from exposure). The same amount of TVCs was detected over time in all experimental conditions. In biofilms exposed to NN (Figure 2A) and NN supplemented with ciprofloxacin (NN + CIP, Figure 2B), the number of culturable cells was stable until T120; this was followed by a constant increase in culturable cells until the end of the experiment, when VBNC cells were about 70%. In tobramycin-exposed (NN + TOB) PAO1-N biofilms, the number of culturable cells was substantially stable up to T135, then it increased slightly; at the end of the experiment, VBNC cells were 97% of TVCs (Figure 2C).These findings indicate the development, in all test conditions, of a VBNC population (calculated as the difference between TVCs and culturable cells), which shrank gradually until the end of the experiment in PAO1-N NN and NN + CIP biofilms—likely due to the replication of culturable cells—whereas in NN + TOB biofilms, it remained fairly stable.

In contrast, at the first time point, *P. aeruginosa* C24 biofilms showed a narrow difference between culturable cells and TVCs, which was followed by a constant reduction in the formers in all test conditions until the end of the experiment. This trend was particularly marked in NN + TOB C24 biofilms (Figure 2F). In NN and NN + CIP C24 biofilms, culturable cells showed a marked reduction only between 90 and 120 days, whereas they remained unchanged from T120 until the end of the experiment (Figure 2D,E). Accordingly, NN and NN + CIP C24 biofilms did not show a stable VBNC population, whereas in NN + TOB C24 biofilms, a significant VBNC subpopulation was documented from T90 until the end of the experiment.

One further difference between the two strains was the appearance of specific colony phenotypes, observed throughout the experiment. Starting from T60, NN + TOB C24 biofilms subcultured on cystine lactose electrolyte deficient (CLED) agar produced slow-growing (i.e., not detectable before 72 h incubation), non-mucoid and non-swarming colonies, whereas NN C24 biofilms produced non-mucoid, fast-growing and swarming colonies. *P. aeruginosa* PAO1-N biofilms, starting from T120, produced a mucoid slow-growing phenotype in all test conditions when subcultured on CLED agar.

### 2.3. Strain-and Antibiotic-Dependent Abundance of VBNC P. aeruginosa Cells

The possible involvement of low-dose antibiotics in the induction of the VBNC phenotype [10] was tested by exposing PAO1-N and C24 biofilms to three stress conditions: starvation, starvation in the presence of 1/4 MIC CIP and starvation in the presence of 1/4 MIC TOB. The VBNC P. aeruginosa subpopulations detected in the three conditions were quantified and compared (Table 1).

A VBNC subpopulation was detected in all PAO1-N biofilms as early as 60 days from stress exposure at a frequency of 85% of TVCs (NN + CIP) or higher (NN and NN + TOB). However, after further 60 days (T120) their count fell to 67.7% in biofilms only starved and to 73.2% in biofilms exposed to 1/4 MIC CIP, whereas in biofilms exposed to 1/4 MIC TOB, the count actually increased to 97.2% (170 days).

At 60 days, VBNC cell abundance was lower in C24 than in PAO1-N biofilms, their highest frequency being 66.1%, observed in NN + TOB biofilms. From this time on, the VBNC C24 subpopulation in NN and NN + CIP biofilms varied among time points, reaching respectively 51.1% and 38.7% of TVCs 170 days from stress exposure. In contrast, and similar to PAO1-N biofilm counts, C24 biofilms exposed to NN + TOB had a more stable (> 60%) VBNC subpopulation, which at 170 days reached 93% of TVCs.

## 3. Discussion

*P. aeruginosa* biofilms are the main cause of recurrent pulmonary infection in CF patients. Although several features of the infectious biofilm lifestyle, including signaling pathways [10,15] and antibiotic resistance mechanisms [16], have largely been elucidated, little is known about the factors involved in the development of persistent and dormant bacterial forms. Persisters were initially described as bacterial forms which are produced stochastically and whose low metabolic activity is responsible for the failure of antibiotic treatment [17]; this is particularly critical in immunocompromised subjects such as CF patients [18]. More recently, a distinction between spontaneous and triggered persistence has been made [19]. The VBNC state is a later dormant state of persistence, characterized by the loss of the ability to grow on culture media; it can be induced by a variety of environmental stressors including starvation, excess light, temperature fluctuations and exposure to toxic compounds [5]. Low antibiotic concentrations can also contribute to the trigger of the VBNC state [11,12]. The present study was devised to establish whether starvation and subinhibitory drug concentrations—two typical features of the deepest biofilm layers [2,9]—induced VBNC forms in biofilms of a clinical (C24) and a laboratory (PAO1-N) strain of *P. aeruginosa*.

The *ecfX*-targeting qPCR used in this work has been previously described to accurately quantify TVCs due to the ability of the DNA extraction protocol to exclude eDNA [8]. The present work further supports the reliability of this procedure since *P. aeruginosa* counts obtained by qPCR resulted comparable to the amount of viable cells detected by flow cytometry (Appendix A).

VBNC cells of both *P. aeruginosa* strains were detected in all experimental conditions, in line with our previous study on *S. aureus* biofilms [11] and with the notion that the loss of culturability may be a survival strategy triggered by adverse environmental conditions, including starvation and toxic compounds [20,21,22].

Balaban et al. [19] have made a distinction between spontaneous and triggered persistence and indicated starvation as one of the most common triggers; they reported that when the trigger is removed, the persistent phenotype may be retained [19,23]. VBNC cells possess these features, although the factors that are required for the recovery of culturability are still unclear [11,21]. These data suggest that the VBNC state is a more advanced and less metabolically active state of triggered persisters and agree with the reports describing VBNC cells as the main obstacle to infection eradication [5]. Notably, after culture without nutrients for two months, all viable *P. aeruginosa* cells, either culturable and non-culturable, are to be considered as persisters. However, exposure of culturable persisters to a nutrient-rich environment can give rise to an antibiotic-susceptible subpopulation with full metabolic activity, whereas VBNC cells require the presence of specific resuscitation factors [21]. At variance with *S. aureus* biofilms exposed to vancomycin or quinupristin/dalfopristin [11], which uniformly lost culturability, a subset of culturable cells was detected in all *P. aeruginosa* biofilms. This different behavior may be explained by the greater stress-response ability of the opportunistic pathogen *P. aeruginosa,* which is endowed with a larger genome encoding several metabolic pathways and regulation systems to face the adverse environmental conditions. Among these, the lack of nutrients may be overcome via autophagy, with a small bacterial population feeding on lysed kin cells [24] and, after proliferation, restoring the original cell abundance.

Although VBNC cells were detected in both *P. aeruginosa* strains in all conditions, they were more numerous in biofilms exposed to subinhibitory tobramycin concentrations, as highlighted by their much greater prevalence in the tobramycin-exposed biofilms of both strains starting from T135 (Table 1). Subinhibitory tobramycin seems to hinder reversion to the culturable state or replication of the culturable subpopulation still found in the biofilm. This is suggestive of a hormetic response, i.e., a biphasic dose–response relationship (specifically, low-dose stimulation and high-dose inhibition) described by Davies and co-workers for various antibiotics, including aminoglycosides [10]. These results are also in agreement with those reported by Carvalhais et al., who described a role for protein synthesis inhibitors in the development of VBNC *S. epidermidis* [25], and by Lee and Bae for *P. aeruginosa* [12].

However, the two strains used in this study showed different behaviors. Whereas TVCs declined continuously in all C24 biofilms, their amount in tobramycin-exposed PAO1-N biofilms remained unchanged and slightly increased at the end of the experiment, always reaching greater values than those detected in C24 biofilms. This suggests a strain-specific behavior reflecting the greater ability of the lab strain to adjust to unfavorable conditions compared to the CF strain, which is used to living in symbiosis with the host. The more limited viability of the latter strain could be explained either by the maintenance in a non-symbiotic condition (in vitro biofilms) or by the loss of genetic information due to adaptation to the lung environment. Moreover, the mucoid shift detected in *P. aeruginosa* PAO1-N after stress exposure for 120 days is likely capable of protecting the bacterial cells, thus favoring strain survival.

## 4. Materials and Methods

### 4.1. Bacterial Strains, Growth Media and Chemicals

The extensively drug-resistant strain *P. aeruginosa* C24, isolated from a sputum sample of a chronic CF patient, and the reference strain *P. aeruginosa* PAO1-N, kindly provided by Prof. Paul Williams (Centre of Biomolecular Sciences, University of Nottingham, Nottingham, UK), were used as models. *P. aeruginosa* ATCC 27853, from the strain collection of the Microbiology section of the DiSVA (Polytechnic University of Marche, Ancona, Italy), was the reference strain in antibiotic susceptibility tests. All strains were cultured in lysogenic broth (LB) or CLED agar plates (all from Oxoid, Thermo Fisher Scientific, Waltham, MA, USA). Ciprofloxacin and tobramycin were purchased from Sigma-Aldrich (Saint Louis, MO, USA).

### 4.2. Antibiotic Susceptibility Test

The MIC of ciprofloxacin and tobramycin was determined by the broth microdilution method according to CLSI guidelines [26] using *P. aeruginosa* ATCC 27853 as the reference strain.

### 4.3. In Vitro P. aeruginosa Biofilm Development and Monitoring

Biofilm models were developed in 35 mm petri dishes as described previously [27], with some modifications. Briefly, 2 mL aliquots of exponential *P. aeruginosa* cultures grown in LB broth were diluted to an optical density (OD) of 600 = 0.1, inoculated into 35 mm petri dishes and grown at 37 °C for 48 h, refreshing the medium after the first 24 h. Next, the medium was discarded and replaced with 3 mL of NN broth (M9 minimal medium without glucose) [11], alone or supplemented with 1/4 MIC of ciprofloxacin or tobramycin. This antibiotic concentration was tentatively chosen with reference to that used for *S. aureus* by Pasquaroli et al. [28]. The NN medium without/with antibiotic was refreshed every week. The abundance of culturable and TVCs, i.e., culturable and non-culturable, was assessed at 60, 75, 90, 120, 135, 150 and 170 days, considering the first day of incubation in NN broth (starvation) as time (T) 0. VBNC cells abundance was calculated as the difference between TVCs and culturable cells. At each time point, 3 biofilms per experimental condition were mechanically detached, resuspended in 1 mL phosphate buffered saline (PBS) and serially diluted 10-fold in PBS. Suitable dilutions were used in subsequent analyses.

### 4.4. Biofilm Quantification

Biofilm production and removal was measured in both *P. aeruginosa* strains by crystal violet staining as described previously [29], with some modifications. Briefly, *P. aeruginosa* biofilms were developed in 35 mm petri dishes in LB broth. After overnight incubation at 37 °C, the planktonic phase was removed, and its OD was recorded at 600 nm. Biofilms were mechanically detached; after removal, the matrix was quantified by optical density (OD_600_). The sessile phase was washed with sterile deionized water (DW) and stained with 1% crystal violet for 15 min. After removing the dye, biofilms were washed with DW, resuspended in 96% ethanol and quantified by measuring OD at 570 nm. Biofilm production was normalized by calculating the OD_570_/OD_600_ ratio. These tests were run in triplicate; each strain was tested for the amount of biofilm production and detachment.

### 4.5. P. aeruginosa CFUs, qPCR and Flow Cytometry Counts

The abundance of total viable and culturable *P. aeruginosa* cells was determined by a species-specific qPCR and plate count, respectively.

For plate counts, serial dilutions of *P. aeruginosa* biofilms were plated on CLED agar; the abundance of CFUs was assessed after incubation for 24, 48, and 72 h at 37 °C.

For qPCR counts, 1:100 dilutions of 1 mL-PBS-resuspended *P. aeruginosa* biofilms were treated with DNase (reported to result as effective as propidium-monoazide in excluding the interference of the eDNA by Reyneke and colleagues [30]) and total DNA was extracted using QiAamp DNA Mini Kit (Qiagen, Hilden, Germany). Two µL of DNA were then amplified using the previously developed *ecfX*-targeting qPCR [8].

qPCR counts were validated by FC assays after live/dead staining using 1X SYBR green and 40 mg/L propidium iodide [31]. Assays were performed in a Guava Millipore cytometer using 200 µL of a 1:1000 dilution of *P. aeruginosa* biofilms and analyzed by GUAVASOFT 2.2.3 software. Side scatter and green (GRN) fluorescence were gated to discriminate bacterial cells from the background; both channels were used at 488 nm and a threshold value was set in the GRN channel. SYBR green and propidium iodide were excited using a 488 nm laser, and the emissions were collected at 525/30 and 617/30 nm, respectively. Signal detection was enhanced by logarithmic amplification (4 decades); to increase statistical significance, the total number of particles analyzed was set at 20,000 events/replicate. All assays were run in technical duplicate and biological triplicate.

A difference ≥ 0.5 log between plate and qPCR/FC counts was held to indicate the presence of a VBNC *P. aeruginosa* subpopulation [8].

### 4.6. Statistical Analysis

The significance of the difference between qPCR and plate counts was assessed by Student’s t-test (threshold, 0.05).

## 5. Conclusions

In conclusion, our findings highlight a strain-specific response to stress conditions and suggest that tobramycin, which is routinely used to treat CF lung infection, might contribute to the induction of VBNC *P. aeruginosa* forms in the lung of CF patients, likely through its ribosome binding-ability, which might trigger bacterial gene pathways involved in the development of the non-culturable phenotype [32].

They also stress the need for the routine adoption of a non-cultural microbiological approach to the diagnosis of *P. aeruginosa* pulmonary infection and confirm the usefulness of qPCR [8,33]; moreover, they suggest that the development of a species-specific FC protocol could constitute an alternative routine diagnostic technique.

The involvement of tobramycin in VBNC cells abundance in *P. aeruginosa* biofilms warrants further investigations, including a greater number of strains and the behavior of other drugs, particularly those included in the therapeutic protocols for CF *P. aeruginosa* lung infection.

## Figures and Tables

**Figure 1 antibiotics-09-00399-f001:**
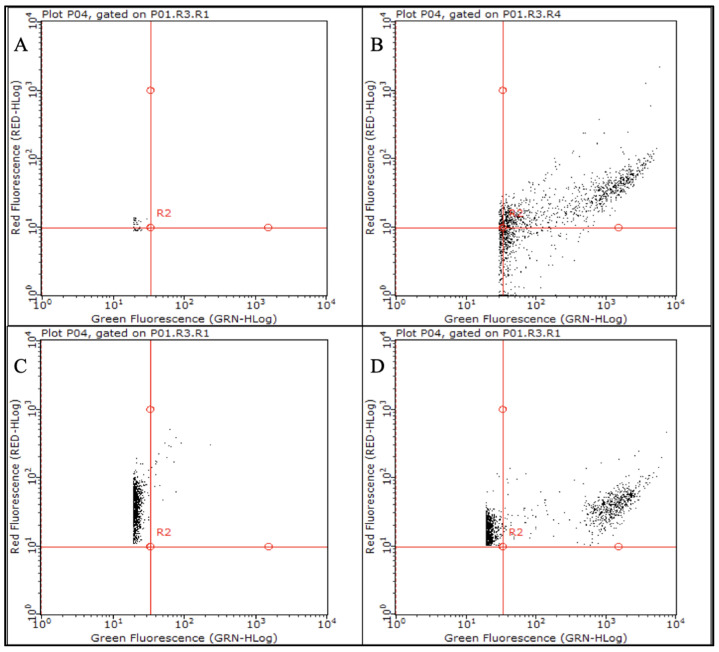
Analysis of a *Pseudomonas aeruginosa* PAO1-N biofilm by flow cytometry. The biofilm was analyzed after 120-day exposure to non-nutrient (NN) + tobramycin to determine its content in total viable cells, either undiluted or diluted 1:1000 in PBS. (**A**) PBS, negative control; (**B**) SYBR green-stained diluted biofilm; (**C**) Propidium iodide-stained diluted biofilm; (**D**) Live/dead-stained diluted biofilm. Viable cells were SYBR green- and live/dead-stained, whereas dead cells were propidium-iodide-stained.

**Figure 2 antibiotics-09-00399-f002:**
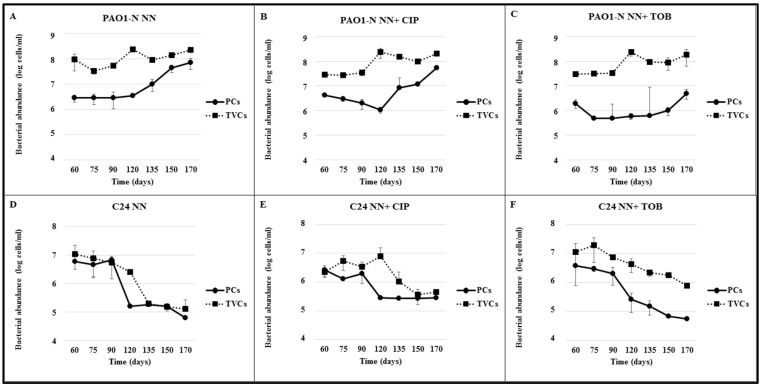
Viable cell counts determined in *P. aeruginosa* PAO1-N and C24 biofilms at different time intervals from stress exposure. Biofilms of *P. aeruginosa* PAO1-N and C24 were developed in vitro for 48 h in rich medium and subcultured in: (**A**,**D**) absence of nutrients (NN); (**B**,**E**) NN supplemented with CIP (NN + CIP); and (**C**,**F**) NN supplemented with TOB (NN + TOB). Total viable cells (TVCs, calculated as the average of qPCR and flow cytometry counts) and culturable cell counts at T0 were 1–2 × 10^9^/mL (C24) and 1–4 × 10^9^/mL (PAO1-N). Solid circles: culturable cells, detected by plate counts (PCs); solid squares: TVCs, culturable and non-culturable cells determined by qPCR. All counts are the average of three biological replicates ± standard deviation. CIP, ciprofloxacin; TOB, tobramycin.

**Table 1 antibiotics-09-00399-t001:** VBNC *P. aeruginosa* abundance in the PAO1-N and C24 in vitro biofilms exposed to starvation alone or starvation in the presence of subinhibitory antibiotic concentrations.

Strain	Time (Days)	TVCs (VBNC%)		
		NN	NN + CIP	NN + TOB
PAO1-N	60	9.51 × 10^7^ ± 0.13 (97)	2.85 × 10^7^ ± 0.23 (85.1)	2.98 × 10^7^ ± 0.14 (93.4)
	75	3.41 × 10^7^ ± 0.84 (91.6)	2.76 × 10^7^ ± 0.48 (89.1)	3.18 × 10^7^ ± 0.50 (98.5 *)
	90	5.39 × 10^7^ ± 1.06 (91)	3.46 × 10^7^ ± 0.80 (94.2)	3.29 × 10^7^ ± 0.04(98.5)
	120	2.45 × 10^8^ ± 0.56 (98.6)	2.37 × 10^8^ ± 0.06 (99.5)	2.32 × 10^8^ ± 0.75 (99.7 **)
	135	9.43 × 10^7^ ± 0.00 (89)	1.53 × 10^8^ ± 0.00 (88)	9.20 × 10^7^ ± 0.00 (99.3 *)
	150	1.46 × 10^8^ ± 0.00 (70)	9.86 × 10^7^ ± 0.00 (88.1 **)	8.98 × 10^7^ ± 1.73 (98.8 **)
	170	2.26 × 10^8^ ± 0.00 (67.7)	2.05 × 10^8^ ± 0.00 (73.2)	1.83 × 10^8^ ± 0.63 (97.2 **)
C24	60	1.07 × 10^7^ ± 0.09 (44.8)	2.21 × 10^6^ ± 0.73 (ND)	1.12 × 10^7^ ± 0.01 (66.1)
	75	7.68 × 10^6^ ± 0.53 (69.7)	5.31 × 10^6^ ± 1.06 (75.5)	1.96 × 10^7^ ± 0.02 (84.7 **)
	90	5.35 × 10^6^ ± 0.00 (ND)	3.35 × 10^6^ ± 0.86 (40.3)	7.35 × 10^6^ ± 0.07(72.1 **)
	120	2.49 × 10^6^ ± 0.30 (93.8)	7.85 × 10^6^ ± 0.45 (96.4 **)	4.29 × 10^6^ ± 2.10 (94 **)
	135	1.93 × 10^5^ ± 0.31 (ND)	1.05 × 10^6^ ± 0.00 (72.4 **)	2.21 × 10^6^ ± 0.55 (93.2 **)
	150	1.51 × 10^5^ ± 0.44 (ND)	3.58 × 10^5^ ± 0.79 (23.2 **)	1.80 × 10^6^ ± 0.06 (62.1 **)
	170	1.27 × 10^5^ ± 0.06 (51.1)	4.57 × 10^5^ ± 0.33 (38.7)	7.84 × 10^5^ ± 0.61 (93 **)

* *p* ≤ 0.05, ** *p* ≤ 0.001; ND, not detectable; TVCs, total viable cells/mL expressed as average of three biological replicates ± standard deviation.

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
