# Peer review of "Role of Tobramycin in the Induction and Maintenance of Viable but Non-Culturable Pseudomonas aeruginosa in an In Vitro Biofilm Model"

_antibiotics, 2020, doi:10.3390/antibiotics9070399_

Round 1

Reviewer 1 Report

Authors in the manuscript “Role of tobramycin in the induction and maintenance of Viable but Non-Culturable Pseudomonas aeruginosa in an in vitro biofilm model” determine the effect of low tobramycin concentrations inducing the VBNC phenotype which could contribute to recurrence of Pseudomonas aeruginosa infections.

Lines 18, 38-39, 165-168, and other. “A pivotal role is played by the occurrence of antibiotic-unresponsive persisters, including viable but non-culturable (VBNC) forms”, “Viable but non-culturable (VBNC) cells have been described as a late stage of antibiotic-persisters”, and similar.

I disagree with the interpretation that authors make of persister and VBNC states because several studies describe two well-defined and closely related dormancy states in nonsporulating bacteria: antibiotic persistence and the viable but nonculturable state (VBNC).  Antibiotic persisters are described as a subpopulation of cells that is tolerant to antibiotics even though the majority of the population had succumbed to the treatment. The VBNC state describes a subpopulation of bacteria that have transiently lost their ability to grow on routine laboratory media on which they were previously able to grow. They are nondividing and, unlike persisters, are unable to immediately regain the ability to divide when plated on routine laboratory medium.

Balaban NQ, Helaine S, Lewis K, Ackermann M, Aldridge B, Andersson DI, Brynildsen MP, Bumann D, Camilli A, Collins JJ, Dehio C, Fortune S, Ghigo JM, Hardt WD, Harms A, Heinemann M, Hung DT, Jenal U, Levin BR, Michiels J, Storz G, Tan MW, Tenson T, Van Melderen L, Zinkernagel A. 2019. Definitions and guidelines for research on antibiotic persistence. Nat Rev Microbiol: 17:441-448. Doi: 10.1038/s41579-019-0196-3

Ayrapetyan M, Williams T, Oliver JD. 2018. Relationship between the viable but nonculturable state and antibiotic persister cells. J Bacteriol 200:e00249-18. Doi: 10.1128/JB.00249-18

Ayrapetyan M, Williams TC, Baxter R, Oliver JD. 2015. Viable but nonculturable and persister cells coexist stochastically and are induced by human serum. Infect Immun 83:4194 –4203. doi:10.1128/IAI.00404-15

Ayrapetyan M, Williams TC, Oliver JD. 2015. Bridging the gap between viable but non-culturable and antibiotic persistent bacteria. Trends Microbiol 23:7-13. Doi:10.1016/j.tim.2014.09.004

Li L, Mendis N, Trigui H, Oliver JD, Faucher SP. 2014. The importance of the viable but non-culturable state in human bacterial pathogens. Front Microbiol 5:258. Doi:10.3389/fmicb.2014.00258

Lewis K. 2006. Persister cells, dormancy and infectious disease. Nat Rev Microbiol 5:48–56. Doi:10.1038/nrmicro1557

I think the introduction and discussion should be rewritten considering this perspective.

Line 55. Italics for P. aeruginosa

Reviewer 2 Report

In the manuscript by Mangiaterra et al, the authors suggest that tobramycin induces and maintains viable but non-culturable (VBNC) Pseudomonas  aeruginosa in an in vitro biofilm model through a stress response.  The underlying premise is that growing the bacteria at ¼ MIC induces this stress response.  However, they used starvation conditions prior to this treatment, and this should induce the bacteria to become VBNC.  Therefore, it is unclear how the additional use of antibiotics can impact this response.  In addition, they provide no information to support the choice of this antibiotic concentration as appropriate, either thought data in the manuscript or to previously published work.  Since the authors did not do any comparisons in non-starvation conditions, they do not know what impact the exposure to antibiotics has as a stress inducer. The data in figure 2 might suggest that the addition of antibiotic has no effect on a given strain.  Finally, drawing broad conclusions from the analysis of two strains is problematic.

Specific comments

Overall, it is difficult to see how the data contained in the manuscript supports the authors’ conclusions. The question each figure is addressing is not clearly described, and the important points from each figure are not articulated.  This can be illustrated in their conclusion in line 89  “However, the two strains showed different behaviours.”  I am not sure to what behavior measurement they are referring. Later they state, “exhibited a longer lag phase.”  What does this mean?  Overall, the data contained in the figures represent a decrease from the original inoculum.  Can this just be stochastic fluctuation?  The author’s conclusions need statistical support

The authors have not done any type of biofilm quantification, and they do not provide any evidence for potential differences in biofilm removal between the two strains.  Hence it is difficult to compare results between two different strains.  I think what they are trying to say in Table 1 is a quantification of the data in Figure 2, where the terminal readings suggest that there are more nonviable cells in the presence of tobramycin at day 170.  However, this is not discussed.  They also do not address the scientific rigor.  How many times was this experiment performed?  How repeatable are the data.  From the methods section, it appears that this is a single experiment performed once in triplicate.

Reviewer 3 Report

This manuscript describes the existence of viable but non-culturable cells in Pseudomonas aeruginosa biofilms in relation to sub-inhibitory levels of tobramycin and ciprofloxacin, two frequently used antibiotics for lung infections in CF patients. This an important topic as CF patient treatment of chronic infection with this organism is very challenging. The paper is clearly written, the experimentation is appropriate, and the results are well-described.

Minor comments:

Line 27-this sentence does not make much sense and needs to be changed.

Line 37-change “districts” to “regions/pockets of the lung”

Line 55-P. aeruginosa-needs to be italisised.

Line 76: Please define “FC” as “Flow cytometry” here.

Line 102: For PAO1-N biofilms, rather than “PAO1-N biofilms”?

Line 190: This sentence needs to be re-phrased. Perhaps the following would make more sense: “This suggests the latter is mainly responsible for the lack of successful infection eradication.”

Line 207: “suggests”, rather than “suggets”

Line 208: “used to living”.

Lines 208-209: The authors discuss the differences between C24 and PAO1-N but do not comment more widely on why this might be the case. It is unexpected that the laboratory isolate would adapt more successfully than the CF isolate. Perhaps the different colony types produced by C24 in different conditions may be worth elaborating on here? Do the authors perform any proteomic/transcriptomics experiments on any of these colony types? Do they plan to perform these studies with a greater selection of CF isolates from early/chronic infections?

Round 2

Reviewer 1 Report

Line 58. P. aeruginosa cells (space)

Figure 1. The quality of the image should be improved.

Line 100. “1-2x109/ml (C24) and 1-4 x109/ml (PAO1-N)” should be expressed as 1-2x109/ml (C24) and 1-4 x109/ml (PAO1-N)

Figure 2. The quality of the image should be improved. The lines and axes of the graphs are not clear. Expressing the data as 1.00 E+5 is misleading. Since it is clearly explained that bacterial densities are expressed as log, the values should be 1, 2, 3 .---

Line 109. “(about 5x106)” should be expressed as “(about 5x106)”

Table 1. Are all counts and percentages average of three biological replicates ± standard deviation ? DS values should be included.

Line 148. Is “TVCs, log of Total viable cells /ml” correct? The data shown are number cells / ml, no log of number cells/ml.

Lines 186-191. Why are there words in bold?

Line 207. S. epidermidis italic.

Reviewer 2 Report

Many of the comments in the authors rebuttal are problematic.

  1. The authors’ state, “……..starvation is sufficient to induce the VBNC state”. This brings into question the underlying validity of all of their data.
  2. “The concentration of 1/4x MIC was selected because while not inhibiting bacterial growth it is strong enough to elicit a bacterial response (Pasquaroli et al., 2018 doi: 10.1093/jac/dky338)” This paper did not use Tobramycin, and without data, their justification is lacking.
  3. While I appreciate the amount of work required to use a large number of strains, this is needed to provide any scientific rigor to the work.
  4. “We agree that “this is a single experiment performed once in triplicate”;” This is not acceptable for drawing broad conclusions.

Round 3

Reviewer 2 Report

none